# Antioxidative Role of Heterophagy, Autophagy, and Mitophagy in the Retina and Their Association with the Age-Related Macular Degeneration (AMD) Etiopathogenesis

**DOI:** 10.3390/antiox12071368

**Published:** 2023-06-29

**Authors:** Małgorzata Nita, Andrzej Grzybowski

**Affiliations:** 1Domestic and Specialized Medicine Centre “Dilmed”, 40-231 Katowice, Poland; 2Institute for Research in Ophthalmology, Foundation for Ophthalmology Development, Gorczyczewskiego 2/3, 61-553 Poznań, Poland

**Keywords:** age-related macular degeneration, RPE cells, reactive oxygen species, oxidative stress, heterophagy, autophagy, mitophagy

## Abstract

Age-related macular degeneration (AMD), an oxidative stress-linked neurodegenerative disease, leads to irreversible damage of the central retina and severe visual impairment. Advanced age and the long-standing influence of oxidative stress and oxidative cellular damage play crucial roles in AMD etiopathogenesis. Many authors emphasize the role of heterophagy, autophagy, and mitophagy in maintaining homeostasis in the retina. Relevantly modifying the activity of both macroautophagy and mitophagy pathways represents one of the new therapeutic strategies in AMD. Our review provides an overview of the antioxidative roles of heterophagy, autophagy, and mitophagy and presents associations between dysregulations of these molecular mechanisms and AMD etiopathogenesis. The authors performed an extensive analysis of the literature, employing PubMed and Google Scholar, complying with the 2013–2023 period, and using the following keywords: age-related macular degeneration, RPE cells, reactive oxygen species, oxidative stress, heterophagy, autophagy, and mitophagy. Heterophagy, autophagy, and mitophagy play antioxidative roles in the retina; however, they become sluggish and dysregulated with age and contribute to AMD development and progression. In the retina, antioxidative roles also play in RPE cells, NFE2L2 and PGC-1α proteins, NFE2L2/PGC-1α/ARE signaling cascade, Nrf2 factor, p62/SQSTM1/Keap1-Nrf2/ARE pathway, circulating miRNAs, and Yttrium oxide nanoparticles performed experimentally in animal studies.

## 1. Introduction

As the world’s population ages, several chronic, oxidative stress-linked neurodegenerative diseases associated with aging, including Alzheimer’s disease, Parkinson’s’ disease, Huntington’s disease, motor neuron diseases, and age-related macular degeneration (AMD), are becoming increasingly common [1]. AMD is the leading cause of irreversible and severe visual impairment (legal blindness) in patients aged above 65 years in industrialized countries in Europe and North America [2,3], which exerts a strong impact on the daily activities of elderly patients and declines their quality of life [4]. 

AMD affects the central part of the retina (macula), which represents an anatomically and functionally particular tissue complex composed of photoreceptors, retinal pigment epithelium (RPE), Bruch’s membrane (BrM), i.e., extracellular matrix, and choriocapillaries. Visible clinically soft drusen (SD), i.e., deposits beneath RPE and in the BrM formed by accumulation of metabolic waste products consisting of oxidized lipoprotein modifications, as well as pigmentary abnormalities (black lipofuscin granules), loss of postmitotic RPE cells and photoreceptors, and choriocapillaris dropout, characterize early AMD form and are a hallmark of this disease [2,3]. Visual loss in the course of two advanced and terminal AMD stages is caused by the “geographic” atrophic death of the photoreceptors/RPE/choriocapillaris complex, so-called GA/AMD (nonexudative or “dry” AMD), or by the formation of the choroidal neovascular (CNV) membrane as a result of pathological angiogenesis below the retina, so-called CNV/AMD or exudative AMD [2,3]. There has been no effective treatment found for GA/AMD yet. Application of anti-VEGF agents represents a current therapeutic method for exudative AMD. Intraocular injections of anti-VEGF antibodies or VEGF-binding peptides restrain the proliferation of pathological blood vessels and alleviate the symptoms of CNV/AMD [5,6].

AMD is caused by the interactions between the patient’s polygenic background and numerous acquired risk factors. 

Besides non-modifiable risk factors such as genetic background and positive family AMD history, sex (more frequent prevalence in women), and light color of the iris (it acts as a poorer light blocker for the retina than dark iris), important are also multiple modifiable risk factors such as lifestyle, a diet abundant in saturated fatty acids and poor in antioxidants, obesity, hypercholesterolemia, arteriosclerosis, hypertension, smoking [2,3,7,8,9], as well as environmental influences (excessive light, sunlight, and UV exposure) [1,10,11,12] and impact of air pollution. Exposure to air pollutants accelerates or worsens the morbidity and prevalence of AMD since air pollutants change homeostasis, especially macula stress [13].

Molecular mechanisms involved in AMD etiopathogenesis can be broadly categorized into abnormal (excessive) level of reactive oxygen species (ROS) and oxidative stress (OS), generated mainly in the RPE cells [1,10,11], dysregulated internal antioxidant mechanisms such as heterophagy and autophagy pathways [14,15], oxidative mitochondrial dysfunction and sluggish mitophagy pathway [10,16], dysregulated metabolism of lipoproteins [17,18], degeneration of photoreceptors/RPE cells complex and influence of regulated genetically programmed cells’ death (PCD) mechanism leading to massive cells’ death (apoptosis; classical and non-inflammatory PCD pathway, pyroptosis; pro-inflammatory PCD activated by inflammasomes, necroptosis; regulated and caspase-independent necrosis, as well as ferroptosis; novel PCD, all contribute to genetically programmed RPE cell death in AMD) [11,19], chronic inflammation; pathological parainflammation connected with macrophages and microglia recruitment [20,21], complement complex activation [22,23], inflammasomes activation [20,24,25], and dysregulation of angiogenesis [5].

Reactive oxygen species such as hydrogen peroxide (H_2_O_2_), hydroxyl radical (∙OH), or nitric oxide (NO) are permanently produced during normal metabolic reactions in mitochondria. This highly energetic and very unstable redox active species have short half-lives (the cellular half-life of ∙OH is only 10^−9^ s), and therefore, they do not diffuse far from the site of production [26]. Small amounts of ROS are essential for cell signaling and functioning. In physiological conditions, their generation activates cellular antioxidant defense systems, which promote cell survival. However, with time, the imbalance between the antioxidant defense system and excessive ROS production creates oxidative stress in the cellular environment, which disrupts cellular physiology, severely damages multiple cellular processes, activates non-inflammatory PCD in an apoptotic or autophagic mechanism, or induces pro-inflammatory PCD in a pyroptotic manner [26].

Advanced age and its related physiological cell apoptosis and tissue involution, together with genetic predisposition and epigenetic modifications, are the strongest AMD risk factors. A complex mixture of aging, gene influence, lifestyle choices, and environmental factors has an impact on whether this highly non-homogenous disease will develop at all, how rapidly it will advance, and how severe the visual dysfunction ultimately will be [10].

## 2. Heterophagy, Autophagy and Mitophagy Pathways in the Retina

### 2.1. Anti-Oxidative Role of Heterophagy in the RPE Cells

The eye is continuously exposed to various oxidative conditions such as photo-oxidation, ionizing radiation, smoke, and numerous air pollutants. Thus, the light-sensitive retinal tissue, especially the macula, is an area of excessive ROS and oxidative stress creation. Several different physiologic conditions favor the generation of ROS and OS in the retina than in the other tissues of the human body, which include retinal illumination (multi-annual light exposure increases the effect of cumulative OS), the presence of photosensitive molecules (rhodopsin and lipofuscin), and higher concentrations of polyunsaturated fatty acids (PUFAs) in photoreceptors’ outer segments (POS) [16].

The retina is an organ with a higher metabolic and oxygen demand in comparison to other tissues. A higher metabolic rate and a significantly higher amount of blood supply are found in the macula than in the peripheral retina. ROS are generated in the macula as a side effect of physiological retinal metabolism, physiological mitochondrial respiration, and increased oxygen consumption. Moderate ROS amounts are vital to maintaining beneficial cellular processes such as the production of cellular energy and the regulation of mitochondrial protective signaling pathways. However, the combination of a high quantity of PUFAs, high metabolic activities, and high perfusion and oxygenation makes the macula more susceptible to endogenous oxygen, ROS accumulation, and oxidative stress [27].

The vision process initiates photon absorption by the chromophore of visual pigments in photoreceptors (cones and rods). Phototransduction of photoreceptors differs in its signal amplification and inactivation; the cones of the macula serve for detailed and color vision, and the rods of the peripheral retina provide dim light vision. The rods are much more sensitive than the cones; however, the time course of the cones’ photoresponse is about 10 times faster than that of the rods [28,29,30]. Photon absorption leads to isomerization of chromophores (photosensitive molecules) and a phototransduction cascade, which results in photoreceptors’ hyperpolarization and cessation of glutamate secretion at their synaptic terminals. The electric signal generated in photoreceptors is transduced via the optic nerve to the visual cortex in the brain [28,29,30]. Photon absorption and the activation chromophore of rhodopsin (composed of 11-cis retinal, a particular isomer of vitamin A, and opsin, a protein) change the 11-cis retinal configuration into all-trans retinal. Rhodopsin becomes meta-rhodopsin, and its activity is terminated by phosphorylation. Rhodopsin is able to exchange all-trans retinal for 11-cis retinal after additional reaction steps are activated again by a photon. All-trans retinal needs to be re-isomerized to cover a sufficient delivery of 11-cis retinal for proper visual function and needs. Photoreceptors do not express a reisomerase for all-trans retinal. The reisomerization takes place in RPE cells, so all-trans retinal is delivered to the RPE, reisomerized to 11-cis retinal, and delivered back to photoreceptors. The re-isomerization (regeneration) mechanism of the visual pigment chromophore is named the retinoid visual cycle. Re-isomerization takes place mainly in the glial Müller cells for the cones and in the RPE cells for the rods [28,29]. 

Light-sensitive photoreceptors and the RPE cells are continuously exposed to light-induced oxidative stress, stimulated mostly by ultraviolet and blue light. Functional interactions between photoreceptors and RPE cells are essential for visual function. The visual function of RPE cells involves light absorption, participation in the recycling of the retinal visual pigment for the continuity of phototransduction [28,29], and phagocytosis (digestion) of photoreceptors’ outer segments that are permanently destroyed (impair) due to photo-oxidative damage [15]. According to Kaarniranta et al., phagocytosis of exogenous POS material performed by RPE cells is termed heterophagy [15].

Photoreceptors are exposed to constant physiological destruction of their outer segment tips due to photo-oxidative damage and oxidative stress; thus, POS need a constant renewal process from their base to maintain vision. Renewal mechanisms depend on shedding phenomena and persistently performed heterophagy [14,15,31,32]. Phagocytic RPE cells have the ability to engulf and eliminate exfoliated POS [33]. Daily POS heterophagy and degradation of their oxygenated material by the RPE cells lysosomes are necessary for the physiological renewal of photoreceptors, to maintain the normal function of visual cells, and to support retinal homeostasis [14]. The RPE cells are, phagocytically, the most active cells in the whole human body. Daily, they uptake and degrade up to 10% of POS. Each RPE cell is responsible for the phagocytosis of about 30 photoreceptors [34]. Heterophagy is a circadian-regulated process. It takes place in the morning and is triggered by light. Mammals active during the day or night show the same diurnal rhythm of heterophagy. The renovation of the proper (whole) POS length takes approximately eleven days and is maintained by coordinated activity of both POS shedding and heterophagy mechanisms [32].

Heterophagy can be divided into three stages: binding, endocytosis (ingestion), and elimination (digestion). The apical microvilli of RPE cells’ membranes extend around POS, bind to the shed outer segment of the receptors, and create membrane-bound phagosomes. Phagosomes endocytose shedding material intended for elimination (cargo) and transport such internalized biomass by the cytoskeleton and transport vesicles (endosomes) to lysosomes. Phagosomes fuse with lysosomes to form phagolysosomes, and after that, cargo is digested by lysosomes inside phagolysosomes, which finally break down [14,15]. POS material is degraded by lysosomal acid hydrolases. This lysosomal-mediated clearance process is regulated by the acidification of lysosomes with vacuolar-type H + -ATPases (V-ATPase) [35]. Degraded POS material is reused in the photoreceptors. Thus, the efficient activity of the RPE lysosomes is crucial to maintaining normal retinoid levels and the visual cycle [14]. It is not exactly known yet what signaling molecules enable the communication between the RPE and the photoreceptors at the start of phagocytosis; however, the presence of the shed POS seems to be sufficient to initiate this process. Appropriate interplay between different receptors ensures regulation and coordination of the RPE-mediated heterophagy process.

Three molecules, i.e., the CD36 receptor, αVβ5 integrin, and tyrosine-protein kinase c-Mer (MerTK), are involved in the initiation and regulation of heterophagy. The cell surface signaling receptor CD36 is required for both POS internalization (binding) and digestion [36]. αVβ5 integrin is necessary for binding POS to the RPE cells and for initiation of heterophagy in the diurnal rhythm; mice lacking αvβ5 integrin have gradually reduced retinal phagocytic capacity with age via the mechanism by which αvβ5 integrin acts as a POS receiver [37,38], and a deficit or lack of αvβ5 molecule leads to the loss of synchronized circadian heterophagy regulation with a subsequent accumulation of lipofuscin [39]. 

RPE cells recognize and bind phosphatidylserine exposed by POS through MerTK/Gas6 and αVβ5-integrin/MFGE8 interactions. MerTK is a member of the TYRO3/AXL/MER (TAM) receptor kinase family. The MerTK receptor expressed by RPE cells is crucial for POS recognition and endocytosis. TAM ligands bind to and activate Mer kinase to stimulate heterophagy [40]. RPE cells without MerTK cannot engulf the POS, which leads to a complete degeneration of the photoreceptors [41]. Growth-arrest-specific protein 6 (Gas6) is strongly expressed by the RPE cells. It acts as the natural ligand for MerTK and can be recognized as an endogenous, autocrine stimulator of heterophagy [41,42]. Gas6 is the only ligand that has the ability to activate c-Mer, so Gas6 is crucial and required for c-Mer-mediated RPE heterophagy [40]. The glycoprotein milk fat globule-EGF8 (MFG-8) expressed by the RPE cells serves as a ligand and activator of αVβ5 integrin as well as an important activator of diurnal stimulation of phagocytosis in addition to αVβ5 integrin [43]. Activation of the αVβ5-integrin produces an intracellular signaling cascade that involves focal adhesion kinase (FAK). FAK is able to activate MerTK by phosphorylation. Activated MerTK results in an intracellular signaling cascade, which involves the generation of inositol-1,4,5-trisphosphate (InsP3) and a subsequent increase in intracellular free Ca^2+^ and finally promotes heterophagy machinery [44]. The tubby and tubby-like protein1 found in the photoreceptors represent “eat-me” signals directed to RPE cells (and macrophages as well) in order to initiate the heterophagy process, and they are new MerTK ligands for RPE-mediated phagocytosis [45]. 

Figure 1 presents three stages of the heterophagy process.

Photoreceptors’ outer segments with high amounts of PUFAs are particularly susceptible to lipid peroxidation and prone to the accumulation of ROS and oxidative stress. Photoreceptors’ inner segments and RPE cells, both rich with mitochondrial organelles, are also significant sources of oxidative (and nitrative) stress. Unfortunately, along with the growth of age, many factors such as increased ROS activity, decreased efficiency of the heterophagy process, and declining ability of RPE lysosomes all lead to the insufficient digestion of POS and elimination of PUFAs. In turn, oxidated PUFAs accumulate in the form of lipofuscin (called “age pigment”) in RPE lysosomes and photoreceptors’ cell membranes [46]. Lipofuscin, a heterogenous aggregate of proteins, lipids, and carbohydrates, includes toxic fluorophores such as N-retinylidene-N-retinylethanolamine (A2E) and its photoisomers [47]. Lipofuscin is the main photosensitizing factor of the RPE cells [46]. A2E undergoes a variety of photochemical reactions after absorbing a high-energy photon, especially those from 450 to 495 nm (blue light). Photooxidation of A2E accelerates ROS accumulation, generates oxidative stress in mitochondria, and inhibits mitochondrial respiration. Photooxidation of A2E also evokes lipid peroxidation accumulated in cell membranes, which leads to the destabilization of RPE lysosomal membranes and has properties for disrupting other cellular membranes as well [47]. The buildup of autofluorescent granules of lipofuscin causes RPE damage and leads to AMD progression [46,47].

A dysregulated heterophagy pathway in aged and sluggish RPE cells leads to advanced stages of AMD with time in predisposed patients [15].

#### Antioxidative Function of the RPE Cells

RPE is a pigmented cell layer that, at the apical (inner) surface, interacts with photoreceptors’ outer segments and, at the basal (outer) surface, connects with the acellular Bruch’s membrane. BrM conducts signals between RPE cells and underlying choroidal vessels. RPE forms a fundamental component of the retina, which plays an essential role in visual function and in maintaining homeostasis in the microenvironment inside and outside the retina. RPE cells act as the outer blood–retinal barrier between the photoreceptors and choroid; RPE cells form tight junctions, including ZO-1, occluding, and claudin, which function as barriers. RPE cells deliver blood-derived oxygen and nutrients to photoreceptors, release polarized growth factors, and regulate the transport of water, ions, metabolites, and waste products between photoreceptors and choriocapillaris. RPE cells sustain the retinoid visual cycle and are involved in vitamin A metabolism, heat exchange, immunity, and photoreceptors’ and choriocapillaris’ maintenance [48,49]. 

RPE cells function in a highly photo-oxidative microenvironment. They are exposed to photo-oxidative energy and excessive ROS cumulation from both the photoreceptors and the choroid side (overflow of the venous blood and oxygen saturation greater than 90% make the choroid the source of many free radicals). Efficient RPE cells are capable of maintaining the structural integrity of the retina through capable defense systems against light energy, photo-oxidative exposure, and free radicals [29,50]. 

Antioxidants are chemical compounds (molecules) that reduce ROS levels and keep oxidative damage to a minimum. High concentrations of natural non-enzymatic and enzymatic antioxidants in photoreceptors and RPE cells and their ability to repair oxygenized proteins, lipids, and DNA form a defense line [29,50]. Macular pigment, formed by two non-enzymatic dihydrocarotenoids, i.e., lutein and zeaxanthin, is a natural barrier that protects the macula from oxidative damage. Melanosomes and pigment granules move inside the cytoplasm to the apical RPE processes under light conditions. The light absorption by the melanosomes heats up the RPE/choroid complex to temperatures above 40 °C. High perfusion of the choroid also serves to transport away the heat from the retina [29,50].

The macular antioxidant defense also includes other non-enzymatic antioxidants such as glutathione, albumin, bilirubin, uric acid, vitamins A, C, and E, lipoic acid, carotenoids, plant polyphenols, zinc, and selenium, as well as various antioxidative enzymes such as superoxide dismutase (SOD), catalase (CAT), and glutathione peroxidase (GPx). The antioxidant function of RPE cells depends on absorbing light through intracellular melanin and eliminating accumulated ROS through intracellular enzymatic antioxidants, i.e., SOD, CAT, and GPx. In physiological conditions, endogenous antioxidant enzymes reduce almost the total amount of ROS by their direct scavenging. It enables RPE cells to restore their redox status and repair cell damage. CAT is increased in healthy RPE cells during heterophagy to directly scavenge H_2_O_2_ and prevent unnecessary ROS increases [29,50].

A critical balance between ROS production and antioxidant levels is essential to allow for the unrestrained functioning of the RPE cells. Effective heterophagy, the appearance of melanosomes, and natural antioxidants provide protection of the photoreceptor/RPE/choroid complex and maintenance of the structural integrity of this complex for decades [29,50]. The retina is equipped with numerous antioxidant enzymes; however, with age, both the capacity and efficiency of the endogenous antioxidants decrease. The age-related misbalance between the declining potential of the enzymatic antioxidative defense system and excessive ROS production leads to the prevalence of oxidative stress in the RPE cells and the whole retina, which makes the retinal tissue highly vulnerable [51]. With age, long-standing oxidative stress disrupts RPE tight junctions and the retinal–blood barrier, insults the photoreceptors and the PRE cells, leads to intracellular accumulation of detrimental products, stimulates SD formation, and accelerates excessive (even massive) RPE cell death [51]. Along with the aging process, there are also various long-standing, exogenous oxidative impacts such as UV light, cigarette smoke, and many other AMD risk factors [51]. Strong and prolonged (multi-annual) oxidative stress disrupts the normal antioxidative potential of RPE cells, results in irreversible damage to the retina, and finally accelerates AMD progression.

### 2.2. Antioxidative Role of Autophagy in the RPE Cells

Two catabolic pathways, i.e., the ubiquitin–proteasome system (UPS) and autophagy, are functionally connected and cooperate in maintaining proteostasis. Proteostasis means the dynamic regulation of the proteome (the entire set of proteins expressed by a genome) by an integrated network of biological pathways within cells that control the biogenesis, folding, trafficking, and degradation of proteins present within and outside the cell [52]. Smaller and shorter-lived protein substrates are targeted for proteasomal degradation, whereas autophagy machinery is predisposed to degrade and remove larger and longer-lived proteins [53]. 

Autophagy, a cellular housekeeping process also defined as a “self-eating” phenomenon, regulates the homeostasis of proteins, lipids, and nucleic acids, drives mitochondrial quality control, and regulates the amount of ROS and oxidative stress. The term autophagy describes different lysosomal degradation pathways, which act as great cellular self-protective and internal quality control mechanisms involved in the lysosomal removal of cellular waste products (oxidized or misfolded proteins; aggresomes, oxidized lipids, and nucleic acids), as well as faulty cellular components (oxidized or damaged mitochondrial organelles, endoplasmic reticulum (ER), and ribosomes) to maintain the cell’s normal functions and homeostasis. This maintenance molecular pathway directs the lysosomal degradation of unused or damaged cellular molecules and organelles by delivering them to lysosomes, where they are broken down by hydrolytic enzymes. Autophagy improves cell survival during starvation conditions. At this time, increased autophagy acts as a cellular defense and compensatory mechanism that delivers a greater amount of recycling metabolic precursors, i.e., amino acids from strongly degraded polypeptides. Autophagy also plays a pro-survival function during differentiation and development. Moreover, autophagy modulates inflammatory responses and represents an important component of the defense system against different pathogens; it damages protein components of the invasive microbes [52,53,54].

Deregulation of autophagy pathways is involved in many physiological and pathophysiological mechanisms such as cell aging, lysosomal storage diseases, autoimmune diseases, cancer, Parkinson’s disease, Alzheimer’s disease, and others [54,55,56,57] and plays a role in the etiopathogenesis of AMD as well [10,14,15,58,59]. Autophagy, the great cellular clearing process, acts as an essential cellular antioxidant pathway in neurodegenerative diseases [55,58] and represents a potential and promising goal for targeting therapies in both neurodegenerative diseases [54,60] and AMD [10,58,59].

The term autophagy encompasses three main autophagy pathways, i.e., macroautophagy, microautophagy, and chaperone-mediated autophagy (CMA), which differ in how the targeted cytosolic content reaches the lysosomes for degradation [52,54,61,62]. Macroautophagy and CMA are both defined as selective autophagy since they refer to the degradation of specific intracellular components, i.e., aggresomes, lipids, and stress granules, or specific cellular organelles, i.e., mitochondria, ER, and ribosomes [52,54,56]. The terms aggrephagy, lipophagy, granulophagy, mitophagy, reticulophagy, ribophagy, and xenophagy are related to the selective type of autophagy as well and precisely indicate the kind of cargo eliminated, i.e., protein aggregates, lipid droplets, stress granules, mitochondria, endoplasmic reticulum, ribosomes, and exogenous pathogens, respectively [52,54,56]. Non-selective microautophagy occurs under starvation conditions and provides essential nutrients for cell survival [63]. Macroautophagy, an evolutionarily conserved manner of intracellular lysosomal degradation, is the major and best-studied autophagic pathway, which involves five dynamic phases determined as induction, nucleation, elongation, fusion, and degradation. Many (more than 30) autophagy-related genes (Atg) control macroautophagy machinery, and each of its phases is regulated by a relevant group of Atg. Macroautophagy begins with the induction and isolation of a double-membrane pre-autophagosomal structure, i.e., phagophore, in the cytosol from omegasome origin (omegasome is the site from which phagophore, also called isolation membrane, forms). Autophagosome biogenesis is initiated with phagophore nucleation. Cup-shaped phagophore expansion (elongation) allows for the initial engulfing and capturing of large molecules and organelles intended for degradation. Complete closing of the phagophore edges forms and seals an active doble-membrane autophagosome. After autophagosome maturation, its outer membrane fuses with a lysosomal vesicle filled with hydrolytic enzymes and forms the autophagolysosome. Lysosomal degradation of the inner autophagosomal membrane releases hydrolases into the autophagosomal cavity and allows for the degradation of all cargo. Finally, released macromolecules are returned to the cytoplasm and recycled to restore the nutrient balance [52,54].

Autophagosome formation is a regulated process of endomembrane remodeling, guided by membrane factors and autophagy-related proteins in response to cellular stress conditions. Biogenesis of the autophagosome requires membrane acquisition from intracellular compartments. Endomembrane compartments such as the endoplasmic reticulum (ER), the endoplasmic reticulum-Golgi intermediate compartment (ERGIC), the endoplasmic reticulum-exit sites (ERES), endosomes, and plasma membranes create abundant sources of early autophagosomal membrane precursors (vesicles) to form the phagophore and then phagophore elongation to complete the double-membrane autophagosome creation. The phagophore assembly site (PAS) is localized mainly on the ER and harbors the site for autophagosomal membrane assembly, elongation, and completion [64,65,66,67].

A group of proteins named the coat protein complex II (COPII) facilitates the formation of vesicles to transport proteins from the ERES to the ERGIC. Relocation of COPII machinery from the ERES to the ERGIC generates a special type of COPII vesicles, termed the ERGIC-derived COPII vesicles, which serve as a membrane template for the lipidation of soluble microtubule-associated protein 1A/1B-light chain 3 (LC3). LC3 lipidation is a key step in autophagosome biogenesis [66]. The type II transmembrane protein SEC12 (the COPII activator) is normally localized on the ERES, where it governs the relocation of COPII vesicle assembly from the ERES to the ERGIC. In steady-state conditions, SEC12 is enriched in the ERES for protein cargo transport by packaging COPII vesicles. Upon starvation, ERES-SEC12 is enlarged and surrounds the ERGIC, dependent on CTAGE5 and FIP200. CTAGE5; a SEC12 binding protein, is required for the enlargement of the ERES and SEC12 relocation to the ERGIC. FIP200; a subunit of the ULK protein kinase complex, associates with SEC12 via its C-terminal domain and facilitates the starvation-induced enlargement (remodeling) of the ERES-SEC12, independently of the other subunits of this complex (FIP200 is distinct from the ULK/FIP200/ATG13/ATG101 complex on the phagophore assembly site). The remodeling of ERES-SEC12 leads to the relocation of SEC12 to the ERGIC. It depends on the phosphatidylinositol 3-kinase PI3K (PI3K), a crucial element of the PI3K/protein kinase B (Akt)/mTOR signaling pathway, which plays a key regulatory role in autophagy. The relocation of SEC12 to the ERGIC, modulated by P13K, triggers the assembly of ERGIC-COPII vesicles as a membrane template for LC3 lipidation, a potential vesicular pool for the assembly of the autophagosome membrane [66]. A pathway wherein CTAGE5 and FIP200 facilitate starvation-induced remodeling of the ERES, a prerequisite for the production of COPII vesicles budded from the ERGIC, contributes to autophagosome formation [66]. 

Li et al. identified a new type of membrane contact formed between the ERES and the ERGIC generated under conditions of macroautophagy-induction and essential for autophagosome biogenesis [65]. Interactions between the ERES-resident PREB/SEC12 protein and the ERGIC-localized transmembrane p24 trafficking protein 9 (TMED9) establish this new type of membrane contact. The ERES–ERGIC contact mediated by the PREB/SEC12-TMED9 interactions regulates the biogenesis of the ERGIC-COPII vesicle formation, which supports LC3 lipidation and regulates conditions for macroautophagy induction and autophagosome formation. Blockage of PREB/SEC12 relocation from the ERES to the ERGIC only partially affects ERGIC-COPII vesicle formation and autophagosome biogenesis, and transactivation of ERGIC-COPII vesicles from the ERES-localized PREB/SEC12 through the ERGIC-ERES contact likely contributes to a major effect [65]. 

In autophagosome generation, plasma membranes are governed by the Atg9A and Atg16L1 proteins trafficking. Plasma membranes are important sources of Atg9A and Atg16L1 vesicles. Atg9A and Atg16L1 accumulation in the recycling endosomes is an essential step for autophagosome formation [67,68]. Clathrin-mediated endocytosis (CME) is crucial for modulating the protein composition of a cell’s plasma membrane. The adaptor protein complex (AP2) acts as a key adaptor in the CME process. Atg9A and Atg16L1 are both internalized in AP2-clathrin-coated pits at the plasma membranes, and after internalization, Atg9A is delivered to recycling endosomes (RE) through the conventional transferrin receptor internalization route via early endosomes (EE), and Atg16L1 is delivered directly to RE without EE involvement. The heterotypic fusion of Atg9A and Atg16L vesicles in RE is mediated by the vesicle-associated membrane protein 3 (VAMP3) attached to Atg9A-positive vesicles in EE. Finally, Atg9A and Atg16L integrate with the membrane of the nascent phagophore [67]. As the phagophore forms, its membrane surface area expands rapidly while its volume is kept low. The Atg9A protein mediates phagophore membrane expansion. Atg9A and LC3-II are each fully integrated into the expanding phagophore; however, the ratios of these two proteins at different stages of maturation demonstrate that the Atg9 proteins are not continuously integrated but rather are present on the seed vesicles only and become diluted in the expanding autophagosome membrane [69]. Atg9A is the sole multi-spanning transmembrane protein that translocates phospholipids between the outer and inner leaflets of liposomes at the expanding edge of the isolation membrane in the autophagosome-forming machinery. Atg9A colocalizes with the Atg2 protein, which receives phospholipids from the ER. Phospholipids delivered by Atg2 are translocated from the cytoplasmic to the luminal leaflet by Atg9, thereby driving autophagosomal membrane expansion. Both the N- and C-terminal domains of the Atg9A protein are exposed to the cytoplasm. Sorting motifs in the N-terminal cytosolic stretch of the Atg9A protein interact with the adaptor protein AP-2 during the CME process. Mutations in sorting motifs abolish Atg9A’s ability to transport phospholipids between liposome leaflets. It cannot execute autophagy and is abnormally accumulated at the recycling endosomes [68]. 

In contrast to the highly conserved macroautophagy process, chaperone-mediated autophagy, observed only in mammalian cells, is induced by different cell stress conditions such as hypoxia or prolonged cell starvation [52,54]. In chaperone-mediated autophagy, chaperones created by cellular heat shock proteins (HSPs), mainly the heat shock cognate protein of 70 kDa (HSP70, known also as HSPA8), assist in targeting specific substrates for CMA, i.e., proteins containing a KFERQ amino acid motif (Lys-Phe-Glu-Arg-Gln), to lysosomes for enzymatic degradation. CMA substrates containing a KFERQ-like motif are recognized by the HSC70 protein in the cytosol. Then CMA substrates bind to lysosome-associated membrane protein 2A (LAMP2A). The LAMP2A monomer initiates a multimerization process, and the multimeric complex, involving several LAMP2A proteins, forms a translocation cluster through which the CMA substrates are translocated into the lysosomal lumen for degradation [54]. With age, the activity of macroautophagy declines vitally but significantly increases engagement of chaperone-mediated autophagy [70]. Microautophagy depends on the direct engulfing of the cargo by the invaginated lysosomal membrane without forming an autophagosome and represents the simplest mechanism of autophagy [56]. Detailed mechanisms of macroautophagy signaling and chaperone-mediated autophagy have been described in many sources [52,53,54,57]. 

Figure 2 presents the course of macroautophagy, chaperone-mediated autophagy, and microautophagy.

Selective autophagy can be ubiquitin dependent or ubiquitin independent. Ubiquitin is a low-molecular-weight protein that marks proteins intended for degradation, and the addition of ubiquitin molecules to proteins is called ubiquitination. Ubiquitin-conjugated proteins and organelles intended for degradation in the dynamic autophagy process are herded together and sequestered to autophagosomes in the cytosol through the interaction between cargo receptors and autophagosome marker proteins. Macroautophagy and CMA utilize specific autophagosome receptors, i.e., p62 (see below), NBR1 (neighbor of BRCA1 gene 1 protein), NDP52 (nuclear dot protein 52 kDa), and optineurin, to recognize ubiquitin and other structural components on aggresomes or the organelles’ surface in the cargo. The autophagosomes decorated with autophagy-related protein 8/microtubule-associated protein 1A/1B-light chain 3 protein (ATG8/LC3 proteins) are recognized by the LC3-interacting regions (LIR) motif presented on all autophagy receptors. Recognition and chaperoning of cargo by autophagy receptors to the autophagosome is the initial step of macroautophagy. Next, the cargo is chaperoned to the autophagosome, where it is enzymatically degraded. In the ubiquitin-independent mechanism, receptors localized on the damaged organelle, i.e., BNIP3 (member of the apoptotic Bcl-2 protein family) and NIX (also known as BNIP3L), directly link cargo with the LC3 protein on the autophagosome [52,54]. 

p62, also known as sequestosome 1 (SQSTM1); p62/SQSTM1, is a 62 kDa protein that acts as a cargo receptor in p62-mediated autophagy. p62 binds to polyubiquitinated proteins, guides them to the autophagosome, and is sequestered into autophagosome vacuoles. p62 is the main selective autophagy substrate, which links the autophagy pathway and the ubiquitin–proteasome system upon ubiquitinated protein degradation [71]. p62 localizes to the autophagosome formation site on the endoplasmic reticulum. p62 is recruited to preexisting isolation membranes through interaction with LC3, but p62 targeting the autophagosome formation site requires self-oligomerization but not interaction (binding) with LC3 or other autophagy-related globulins (ATGs) [72]. The p62 gene expression can be used as a measure of autophagic flux. Silencing of the p62 gene leads to cargo loading failure and inefficient autophagy, as confirmed by a reduced LC3 conversion ratio, and overexpression of the p62 gene yields the opposite results [73]. p62 is involved in both pro-survival and pro-death cell pathways since it activates the pro-survival TRAF6 (TNF receptor-associated factor 6)-NF-κB pathway [74] and the pro-death-acting polyubiquitinated caspase-8 and a multi-protein complex named DISC (death-inducing signaling complex) [75]. Accumulated proteins p62 are a common component of cytoplasmic inclusions associated with disturbed autophagy and observed in protein aggregations of brain neurodegenerative diseases, including both Parkinson’s and Alzheimer’s diseases [76].

Many molecular regulators have an influence on the activation and course of the autophagy pathway. The mammalian target of rapamycin (mTOR) is the major negative regulator of autophagy. Inactivation of mTOR-dependent mechanisms leads to prompt Atg13 and Atg 101 dephosphorylation, stimulates serine/threonine kinases (ULKs) engagement, promotes FIP-200 protein recruitment, and autophagosome formation [54,57]. mTOR belongs to the PI3K-related kinase family and consists of the central regulatory catalytic core of two functionally distinct multiprotein complexes, i.e., mTORC1 and mTORC2. mTORC1 is the main negative regulator of autophagy [59,60]. Various mTOR-independent mechanisms also have the ability to initiate the autophagy pathway [57]. Adenosinemonophosphate-activated protein kinase (AMPK), a key energy and stress sensor that regulates cellular metabolism, stimulates autophagy by inhibiting mTORC1 [59,60]. Circulating autoantibodies against autophagy regulators compromise the autophagy pathway (see below) [77].

There has not yet been established the unquestioned and direct link between autophagy pathway gene mutations and AMD disease susceptibility. However, polymorphisms or mutations of certain autophagy-related genes may have complicity with the general phenomenon occurring in AMD pathophysiology, and the ADAMTS9 locus is considered a candidate for genetic association between autophagy and CNV/AMD development [78]. 

Under basal conditions, RPE cells rely on protective mechanisms such as UPS. Strong or extended oxidative stress inhibits normal proteasome activity, up-regulates chaperone-mediated autophagy processes, and increases the expression of antioxidant-related genes [79]. The A2E component of lipofuscin is a powerful trigger of the autophagy pathway. In Zang et al.’s study, A2E-treated RPE cells exhibited the formation of autophagosomes after only 15 min of incubation with A2E (25 μM). After 6 h, autophagosomes began to fuse with lysosomes and form autophagolysosomes. The dynamic advancement of the autophagic process, occurring in the RPE cells being simulated by A2E, has also been exhibited by the markers of autophagosome formation, i.e., the buildup of a punctate pattern of cytosolic LC3 and the upregulation of LC3-II and Beclin-1 proteins [46]. An active and efficient autophagy process has an antioxidative and protective role in the RPE cells and maintains the structural and functional integrity of photoreceptors/RPE cells/choroid complex for decades [15]. The HSP70 family, a key effector of CMA [80], and autophagy-related proteins such as LC3 (a marker of autophagy flux) are highly expressed and functionally active in human retinas with senile RPE cells [81]. However, long-standing overwhelmed conditions and upregulated autophagy machinery prompt RPE cell dysfunction and favor SD formation [82]. The impairment course of HSP’s proteolytic pathway in the senescent RPE cells leads with time to dysregulation of the CMA process and, in consequence, to accumulation of oxidative stress-induced damages, build-up of aggregates of oxidized proteins, lipofuscinogenesis, and advancement of AMD pathology [80] in the predisposed patients. 

Under overwhelming conditions, removing aggresomes and cellular protective antioxidant responses in RPE cells involve autophagy machinery mediated by the p62/Keap1-Nrf2 pathway and the expression of antioxidant-related genes; see more in Section 3.2 [1,79,83].

According to Kaarniranta et al., autophagy represents both pro-life and pro-death activity in RPE cells. Pro-life autophagy interactions find expression in the maintenance of homeostasis in postmitotic RPE cells, energy recycling, building material supply, prevention of aggregate formation, degradation of damaged cellular components, and activation of antioxidative Nrf2-dependent defense mechanisms. Self-eating mechanisms and p62-dependent caspase-8 activation prove pro-death autophagy activity [84] since caspase-8 regulates apoptosis [85].

Active mTOR favors detrimental changes in the RPE cells exposed to the impact of oxidative stress; however, rapamycin treatment is capable of preventing these circumstances and preserving the efficiency of the RPE cells and photoreceptors [59,60,86]. The modulation of mTORC1 function is a potential target for the development of therapeutics for neurodegenerative diseases, including retinal diseases. Metformin, resveratrol, a polyphenolic natural compound, and the adenosine analog 5-aminoimidazole-4-carboxamide ribonucleoside (AICAR) activate the AMP-activated protein kinase (AMPK; see more in Section 3.1) and inhibit mTOR. Rapamycin, sirolimus, temsirolimus (CCI-779), everolimus (RAD-001), and deferolimus (AP-23573) are examples of mTORC1 inhibitors as well [59,60].

Rapamycin and A2E act as strong autophagy triggers mediated by the Akt/mTOR pathway; the Akt enzyme is a type of serine/threonine protein kinase, also called protein kinase B, which helps to transfer signals inside cells [46]. In RPE cells being treated (influenced) by A2E, both A2E and rapamycin inhibited the activity of the Akt/mTOR signaling pathway. Moreover, rapamycin ameliorated the detrimental influence of A2E on the RPE cells by inhibiting the inflammatory response and reducing the activity of the pro-angiogenic factor VEGF-A. According to the authors, all these results indicate that appropriate intervention in the autophagy process can be beneficial in restraining AMD development and progression [46]. 

Circulating autoantibodies (AAbs), i.e., anti-S100A9 AAb, anti-ANXA5 AAb, anti-HSPA8, A9, and B4 AAbs, have the ability to impair different autophagy targets and deteriorate autophagy machinery immediately. They may influence the autophagy process indirectly as well as stimulate inflammasomes (cytosolic multiprotein oligomers that promote the secretion of two pro-inflammatory cytokines, interleukin-1β (IL-1β) and interleukin-18 (IL-18), which are responsible for activation of pyroptosis and its interaction with autophagy [76].

S100 calcium-binding protein A9 (S100A9), also known as calgranulin B, triggers both autophagy and apoptosis processes. S100A9 induces translocation of the apoptotic BcL-2 protein (BNIP3) to mitochondria, with subsequent enhancement in ROS production by mitochondria and mitochondrial damage. It in turn increases lysosomal activation within the autophagy (mitophagy) process and apoptotic cell death [87]. The Bruch’s membrane and choriocapillaris of human retinas from AMD patients showed elevated expression of S100 proteins in comparison with normal eyes [88]. Autophagy stimulator Annexin A5 (ANXA5) [89] was documented in BrM and SD [90], and human retinas with CNV/AMD showed increased expression of ANXA5 in comparison with normal eyes [91].

Circulating autoantibodies directed against the HSP70 family disrupt chaperone-mediated autophagy and lead to the accumulation of oxidative stress-induced damage in RPE [80].

### 2.3. Antioxidative Role of Mitophagy in the RPE Cells

Mitochondria are a crucial organelle for cell metabolism. Mitochondria play a role in cellular energetics (they turn the energy from food into energy that the cell can use), stress responses, and various biosynthetic processes. Mitochondria control the regulation of stem cells, innate immunity, and PCD; they contain a self-destructive arsenal of apoptogenic factors to promote cell death. Mitochondria play a role in quickly absorbing calcium ions, which are vital for a number of cellular processes and are involved in cellular Ca^2+^ homeostasis. Mitochondria form the main source of intracellular ROS and superoxide generation, arising as a result and side effect of physiological respiration and cell metabolism, and create the main location of autophagy occurrence [92].

RPE cells’ mitochondria produce important “energy molecules”. The enzyme adenosine triphosphate (ATP) transfers energy to the cells by releasing its phosphate groups. Oxidative phosphorylation of ATP involves the transfer of electrons between multi-subunit complexes (I to IV) of the electron transport chain (ETC) engaged in the reduction of molecular oxygen and creating mitochondrial sources of ROS [93]. Heterophagy, one of the essential daily functions of the RPE, also predisposes these cells to altered mitochondrial redox states and increased mitochondrial DNA (mtDNA) damage [10,14]. About 2–5% of the oxygen is incompletely reduced under normal circumstances [93]. Moderate amounts of ROS are essential to maintain the production of cellular energy and regulate protective signaling pathways in mitochondria [94]. 

Antioxidant enzymes localized in mitochondria, i.e., CAT, manganese superoxide dismutase (MnSOD), peroxiredoxins, and glutathione peroxidases (GPXs), i.e., GPX1 and phospholipid-hydroperoxide GPX (PHGPX), neutralize ROS in a chain of reactions and effectively protect mtDNA from oxidative damage and mutations for a long time. MnSOD catalyzes the dismutation of superoxide radicals into hydrogen peroxide and molecular oxygen. Peroxiredoxins oxidize thioredoxin and remove hydrogen peroxide, which is also decomposed by GPX1 and PHGPX [10,95]. However, over time, DNA repair capacity significantly decreases, supported by a decreased expression of genes encoding 8-oxo-guanine-DNA glycosylase 1 (OGG1), MutY DNA glycosylase, and thymine DNA glycosylase, i.e., enzymes crucial for repairing oxidative mtDNA damage [10]. 

With age, enhanced mitochondrial ROS production and secondarily increased oxidative damage of lipoproteins leads to disruption of mitochondrial structure (calcium ions overload and opening of the mitochondrial permeability transition pore), oxidative damage of the mtDNA genome; the “Common Deletion” and amount of mtDNA mutations [10,96,97]. Dysfunctional mitochondria and their exacerbated generation of ROS are among the earliest events in the progression of numerous neurodegenerative diseases [98]. 

With time, the mitochondria of the RPE cells become more vulnerable to oxidative damage, and their dysfunction is one of the key events in AMD etiopathogenesis [10,99]. A significant increase in mtDNA damage is associated with normal aging, but it is more extensive in AMD disease. mtDNA damage repair and haplogroups (mitochondrial sequence polymorphisms that are found in different populations) are involved in the pathology of this disorder [10]. RPE cells of human AMD donors show changes in select redox proteins, i.e., a significant increase in both CAT and MnSOD. Such upregulation of antioxidant enzymes is comprehended as a compensatory response to the elevated oxidative stress induced by AMD and protection of mtDNA from ROS-induced damage [100].

The maintenance of healthy mitochondria is essential to RPE cells’ homeostasis and survival. Mitophagy, a specialized form of selective macroautophagy, is another antioxidative and pro-survival strategy in RPE cells in response to mitochondrial damage or hypoxic conditions. Dysfunctional mitochondria must be selectively removed from the cell via the mitophagy process to avoid the vicious cycle of excessive ROS production with subsequent mitochondrial damage and ROS release [10,101]. 

Mitochondria form a specific cargo in mitophagy. Mitophagy is a molecular process in which the autophagosome engulfs damaged mitochondria and directs them towards lysosomal degradation. Mitophagy may be dependent on or independent of ubiquitin. PTEN-induced kinase 1 (PINK1), a serine-threonine kinase located on the outer mitochondrial membrane (OMM), and the cytoplasmic E3 ubiquitin ligase (Parkin) are both involved in the removal of dysfunctional mitochondria in a ubiquitin-dependent mitophagy manner [102]. ROS commonly activate the PINK1 Parkin-derived pathway of mitophagy [10].

PINK1 contains a mitochondrial targeting sequence. It occurs in small amounts in properly functioning mitochondria. PINK1 is transported from the outer to the inner mitochondrial membrane, via TOM/TIM membrane translocases located on the outer and inner membrane, respectively, and cut by mitochondrial proteases. The remaining part of PINK1 is released into the cytoplasm, where it undergoes proteolytic degradation [103]. 

Once mitochondria are damaged, the inner membrane depolarizes. It does not always follow the canonical selective autophagy characteristics for the clearance of protein aggregates [10]. The loss of mitochondrial membrane potential is one of the mitophagy-inducing signals [103]. In dysfunctional mitochondria with a loss of membrane potential, PINK1 degradation is inhibited. Thereby, PINK1 binds permanently with the TOM subunit of the TOM/TIM translocases complex and accumulates in the OMM. The accumulation of PINK1 triggers the mitophagy pathway [103]. The inhibition of PINK1 degradation triggers the recruitment of cytoplasmic Parkin [104]. Parkin recruited to the mitochondrial surface is phosphorylated and activated by PINK1. Next, active Parkin initiates the ubiquitination of OMM proteins, including mitofusin-1 and mitofusin-2 (Mfn1 and Mfn2), Miro1, and dynamin-related protein 1 (Drp1) [105,106]. Mitochondria connect to each other and form a spatial and branched network in the cell. Such an organization contributes to the intensification of energy production and facilitates maintaining homeostasis in response to stress conditions. The network can be modified by connecting or disconnecting individual mitochondria depending on the needs [107]. Mfn1 and Mfn2 preserve connections between mitochondria. Decreased activity of Mfn1 and Mfn2 proteins causes the isolation of a dysfunctional mitochondrion from the mitochondrial network. In other words, it prevents the fusion of damaged mitochondrial fragments with healthy mitochondria [105]. The Miro1 protein binds mitochondria to the microtubules and thus provides each mitochondrion with the possibility of movement within the network. Inhibition of Miro1 facilitates isolation and immobilizes the mitochondrion by disconnecting it from the microtubule [106]. Activation of Drp1 by PINK1 and Parkin results in the disconnection of a mitochondrion from the network. Mitochondrial fission, which requires Drp1, is essential for the segregation of damaged mitochondria for degradation [108,109]. Fragments of the damaged mitochondria are escorted by mitophagy receptors to the autophagic machinery, where mitophagy succeeds. 

Mitophagy receptors, i.e., p62/SQSTM1, neighbor of BRCA1 gene 1 protein (NBR1), nuclear domain 10 protein 52 (NDP52), Tax1 binding protein 1 (TAX1BP1), and optineurin (OPTN), contain a ubiquitin-binding domain, which allows their attachment to Parkin-ubiquitinated mitochondria [110]. The adaptor molecule p62/SQSTM1 mediates the aggregation of dysfunctional mitochondria through polymerization via its PB1 domain, in a manner analogous to its aggregation of polyubiquitinated proteins. p/SQSTM1 has a significant impact on early PINK1-dependent mitophagy processes and Parkin-induced mitochondrial clustering; however, it is not essential for the mitophagy process; the mitochondrial voltage-dependent anion channel 1 (VDAC1) is dispensable for both. To put it another way, p62/SQSTM1 is important for mitochondrial function rather than clearance [111]. p62/SQSTM1 depletion leads to altered mitochondrial gene expression and functionality, as well as mitophagy flux. Mutations resulting in the loss of p62/SQSTM1 function and impaired mitophagy lead to neurodegenerative diseases [112].

Polyubiquitin chains attached to the proteins of the OMM recruit proteins binding LC3, which is present on the surface of a maturing autophagosome [113]. When a damaged mitochondrion is attached to an autophagosome, the autophagosome membrane elongates and closes the mitochondrion inside for lysis. Mitophagy is successful [102].

Figure 3 presents ubiquitin-dependent mitophagy.

The inhibitors of ubiquitin-dependent mitophagy (deubiquitinating enzymes), such as USP15, USP30, and USP35, maintain the balance between ubiquitination and deubiquitination. Excessive expression of these enzymes may inhibit mitophagy through the increased removal of polyubiquitin chains [114]. 

Recognition of targeted mitochondria is mediated not only by PINK1/Parkin signaling. Four protein receptors, i.e., Nix, Bcl-2/E1B-19 kDa interacting protein-3 (Bnip3), FUN14 domain-containing protein 1 (FUNDC1), and activating molecule in Beclin1-regulated autophagy (AMBRA1), exist in the outer mitochondrial membrane and contain an LIR sequence recognized by the autophagy machinery. Nix, Bnip3, and FUNDC1 are insignificantly expressed in normoxic conditions; however, they are meaningfully upregulated under hypoxia. Under hypoxic circumstances, selective mitophagic clearance acts as a cellular prosurvival response, which prevents excessive ROS production by the malfunctioning mitochondria [113,115].

The pro-autophagic molecule AMBRA1 has been defined as a novel receptor (activator) of mitophagy in both PINK1/Parkin and p62/SQSTM1-dependent (canonical) and -independent (noncanonical) pathways. In basal conditions, AMBRA1, a Parkin interactor and an upstream autophagy regulator, is present at the mitochondria, where its pro-autophagic activity is inhibited by Bcl-2. After induction of AMBRA1-mediated mitophagy, mitochondria-targeted AMBRA1 induces massive mitophagy in PINK1/Parkin and p62/SQSTM1-competent cells. However, forcing AMBRA1 localization to the outer mitochondrial membrane may change the mitophagy process in a Parkin/p62-independent manner. It is feasible since AMBRA1 has the possibility of binding to the LC3 adapter through an LIR motif, and such interaction is crucial for noncanonical mitophagy, i.e., regardless of Parkin/p62 recruitment. It highlights a novel role for AMBRA1 as a powerful mitophagy regulator, through both canonical and noncanonical systems [116]. Mitophagy depends on the capable and complementary cooperation of both the proteasome and autophagy pathways [113,115].

The novel E3 ubiquitin ligase HUWE1, mostly found in the cytoplasm, collaborates with AMBRA1 in mitochondrial clearance. AMBRA1 regulates mitophagy in two steps. Upon induction of AMBRA1-mediated mitophagy, AMBRA1 favors HUWE1 translocation from the cytosol to the outer membrane of mitochondria. Here, AMBRA1 acts as a cofactor for HUWE1 activity and favors HUWE1 binding to its substrate, i.e., Mnf2 (and probably to other OMM substrates as well). The HUWE1–Mnf2 interaction leads to Mfn2 ubiquitylation and targets them subsequently to the proteasome for enzymatic degradation. In a second step, HUWE1 induces a positive phosphorylation of the LIR motif of the AMBRA1 receptor at the Serine 1014 site, and this modification is mediated by IKKα kinase. Structural changes in AMBRA1 induced post-translationally by HUWE1 and mediated by IKKα increase the binding affinity of the AMBRA1-LIR motif with the LC3 and GABARAP proteins subfamily (mammalian orthologs of the yeast Atg8 protein) and their mitophagic activity. LC3 and GABARAP decorate the autophagosome membrane via its lipidation by a ubiquitin-like conjugation system. Their covalent attachment to lipid membranes is crucial to the growth and closure of the double-membrane autophagosome. HUWE1 is a key inducing factor in the AMBRA1-mediated mitophagy process, which takes place independently of the main mitophagy receptors. HUWE1 promotes PINK1/Parkin-independent mitophagy by regulating AMBRA1 activation via IKKα kinase. And in other words, AMBRA1 regulates mitophagy through a novel pathway, in which HUWE1 and IKKα are the key factors [117]. 

In normal conditions, AMBRA1-dependent mitophagy is inhibited by the highly expressed anti-apoptotic factor MCL1. Overexpression of MCL1 inhibits the mitochondrial recruitment of HUWE1 to mitochondria. MCL1 antagonizes mitochondrial ubiquitylation following AMBRA1 expression by inhibiting mitochondrial translocation of HUWE1, a crucial dynamic partner of AMBRA1, after AMBRA1-mediated mitophagy induction. Thus, MCL1 acts as a stress-sensitive and pro-survival protein in normal conditions, which is able to preserve the mitochondrial membrane from HUWE1 access and delay AMBRA1-dependent mitophagy [118]. However, when AMBRA1-mediated mitophagy is induced by expressing high quantities of AMBRA1 in the mitochondria, MCL1 is phosphorylated by GSK-3β at a conserved GSK-3 phosphorylation site (S159) and degraded by the proteasome through HUWE1 (HUWE1-dependent MCL1 degradation). Hence, MCL1 stability is regulated by the kinase GSK-3β and the E3 ubiquitin ligase HUWE1 in regulating AMBRA1-mediated mitophagy [118].

Figure 4 presents AMBRA1 and HUWE1 interactions in AMBRA1-mediated mitophagy.

Two dynamic mitochondrial processes, fission and fusion, facilitate the constant remodeling of mitochondrial architecture and the mixing of contents from different mitochondrial networks. The coordinated action of fission and fusion segregates dysfunctional mitochondria prior to their selective removal from the cell by mitophagy. Fission leads to mitochondrial fragmentation, whereas fusion results in mitochondrial elongation [10,119]. Fission is one mechanism for ridding the cell of defective mitochondria, whereby damaged segments are sequestered away from the mitochondrial network. Fission followed by selective fusion of adjacent mitochondria occurs under the regulation of the membrane GTPase proteins mitofusin and optic atrophy protein 1. The accumulation of oxidized mitochondrial proteins and reduced respiratory capacity knock down two key proteins of the fission machinery, i.e., dynamin-1-like protein (DRP) and mitochondrial fission 1 protein (FIS1) [119]. Fusion rescues stress by allowing functional mitochondria to complement dysfunctional mitochondria through diffusion and the sharing of components between organelles. Fusion is impaired in mitochondria with reduced membrane potential or those containing damaged mtDNA, thereby isolating the unhealthy mitochondrial fragment and promoting its mitophagic degradation [119]. Failure in any component of fission and fusion machinery may lead to RPE degeneration [119].

Age, oxidative stress, and genetic predisposition are the main risk factors for AMD. The RPE cells, critical for metabolism and homeostasis of the retina, are vulnerable to oxidative stress and other relevant stresses such as high metabolism, high exposure to light, oxidized POS, PUFAs, lipofuscin accumulation, and others, which make them more susceptible to aging and the development of AMD. Heterophagy, autophagy, and mitophagy are regulatory and protective mechanisms intended for the degradation and removal of different cellular components, including those damaged by ROS and oxidative stress, to support cellular renovation and homeostasis [58,120]. However, by definition, heterophagy, autophagy, and mitophagy do not play direct antioxidative roles per se as antioxidants, i.e., natural (or synthetic) enzymatic or non-enzymatic molecules that neutralize free radicals squarely [121]. Antioxidative capabilities diminished with age, impairing heterophagy, autophagy, and mitophagy processes in the RPE cells [120].

ROS play a dual role in modulating the activity of autophagy; they may act as inducers or inhibitors of the autophagy process. ROS trigger the activation of transcription factors such as p62/SQSTM, the nuclear factor erythroid 2-related factor (NFE2L2) (see more in Section 3.1), the hypoxia-inducible factor 1 (HIF-1*α*) (see more in Section 3.3), and forkhead box O-3 (FOXO3) to induce the expression of autophagy-related genes [122]. These transcription factors drive the expression of the genes required for autophagy induction, including BECN1 and LC3, as well as the mitophagy-associated genes, including NIX and BNIP3 [122]. Moreover, ROS oxidize and inactivate Atg4 to maintain lipidated LC3-II and autophagosome formation, and they block PI3K/Akt/mTORC1 signaling to initiate autophagy signaling. On the other hand, the inactivation of autophagy core proteins with sulfhydryl groups sensitive to ROS oxidation, i.e., E1-like enzyme Atg7 and E2-like enzymes Atg10 and Atg3, leads to a reduction in the autophagy process. ROS may inactivate PTEN (a phosphatase) that negatively regulates PI3K/Akt/mTORC1 and diminish autophagy in this manner as well [122]. 

According to Chang et al., ROS may initially oxidize and inactivate essential autophagy genes and then induce several pathways to reactivate autophagy to compensate the redox status, or ROS-modulated autophagy may rely on the context of cell types and the timing or conditions of stress for ROS generation [122]. Autophagy modulated by ROS has several feedback loops to regulate ROS levels. Autophagy degrades ROS-generating organelles (including mitochondria and peroxisomes) during the mitophagy and pexophagy processes; it removes ubiquitinated proteins by binding them to autophagy receptors (p62/SQSTM1, NBR1, and NDP52); it degrades unfolded proteins through chaperone-mediated autophagy; and it activates NFE2L2 to induce antioxidant gene expression to eliminate excessive intracellular ROS [122]. Autophagy plays an antioxidant role in protecting RPE cells from oxidative stress since removing oxidized cellular components eliminates excessive ROS and regulates intracellular ROS levels [122].

Normal aging of the RPE cells relates to a modest decrease in RPE phagocytosis [120,123], impaired autophagy [124,125], a significant increase in mtDNA damage [10], and decreased mitophagy [126]. The development of AMD pathology is related to significantly reduced heterophagy [120,127], dysfunctional autophagy [128], more extensive mtDNA damage [10], significantly decreased mitophagy [101], as well as RPE cell senescence and death [120,129]. Cell senescence, i.e., the state of permanent cellular division arrest, concerns only mitotic cells, connects with aging, and is involved in the pathogenesis of many age-related diseases. Postmitotic RPE cells, although quiescent in the retina, can proliferate in vitro and may undergo oxidative stress-induced senescence [130]. Therefore, according to Blasiak et al., RPE cellular senescence can be considered an important molecular pathway of AMD pathology, which leads to the inability of the macula to regenerate after degeneration of RPE cells caused by factors inducing DNA damage and the autophagy process [130]. 

Autophagy plays a dual role in protecting against retinal degenerative diseases. Enhancement of the autophagy process can mitigate oxidative stress, alleviate oxidative damage, and protect the photoreceptors and RPE cells from degeneration and death [58]. However, overactivated autophagy may lead to retinal cell death [58,131], particularly excessive autophagy at the early stages of retinal diseases [58,132]. Many types of programmed cell death (apoptosis, pyroptosis, necroptosis, and ferroptosis) exist in the RPE cells [11,19,120]. Autophagy regulates the death of retinal pigment epithelium cells in the course of AMD [84]. Autophagy generates caspase-3-mediated cell death (apoptosis) in the RPE cells when they are exposed to excessive oxidative stress [122]. Silencing the autophagy genes Atg5 and Atg7 diminished the death of human ARPE-19 cells treated with H_2_O_2_ in laboratory conditions [133]. Impairing autophagy in the retinal pigment epithelium may lead to inflammasome activation and, finally, pyroptotic RPE cell death. Impairing autophagy may enhance macrophage-mediated angiogenesis and contribute to CNV/AMD development [134]. Thus, appropriate stimulation of autophagy machinery and defining the precise dynamic role of autophagy in the pathogenesis of AMD are essential to choosing optimal time points for retinal neuroprotection [58]. The crosstalk among the p62, NFE2L2, PGC-1, AMPK factors, and PI3K/Akt/mTOR pathway may play a crucial role in improving disturbed autophagy, enhancing autophagy to prevent oxidative injury, and mitigating AMD progression [58]. Mitochondrial dynamic balance and regulation of mitochondrial homeostasis counteract cell death mechanisms. Therefore, mitophagy represents a novel therapeutic target in the prevention or treatment of AMD (atrophic form) as well [101].

## 3. Others Antioxidative Regulators in the RPE Cells

### 3.1. Antioxidative Role of NFE2L2 and PGC-1α Proteins, and NFE2L2/PGC-1α/ARE Signalling Cascade

AMPK, the AMP-activated protein kinase, is a multifunctional protein in cellular metabolism and a crucial energy sensor. The AMPK signaling pathway coordinates cell metabolism, cell growth, apoptosis, and autophagy. It plays a pivotal role in regulating cellular energy homeostasis and cycloprotection (cell survival under stress conditions). Lack of oxygen content, glucose deprivation, lower intracellular ATP levels, and oxidative stress are essential stimulators for AMPK activation [135]. AMPK has a multipurpose role in the protection of mitochondria. It can stimulate mitochondrial biogenesis by increasing the gene transcription regulated by PGC-1α (see below) and can acutely trigger the destruction of existing defective mitochondria through ULK1-dependent mitophagy. Moreover, AMPK is the key player in activating the NFE2L2 signaling cascade (see below) and antioxidant production. Thus, AMPK has the net effect of replacing existing defective mitochondria with new functional mitochondria. The agonist of AMPK has cytoprotective effects by reducing detrimental protein aggregation in RPE cells [136].

AMPK is phosphorylated (activated) in the cytosol in response to oxidative stress. Active AMPK phosphorylates subsequently two proteins accumulated in the cytosol, i.e., NFEL2L and PGC-1α, leading to their activation and nuclear translocation. Active NFEL2L and PGC-1α proteins regulate the production of antioxidant genes and the biogenesis of mitochondria in the nucleus [10]. Failed AMPK-mediated signaling leads to disturbed proteostasis and cycloprotection in RPE cells [10]. AMPK is a putative pharmacological target in the prevention of neurodegenerative diseases and in AMD [137,138].

The nuclear factor erythroid 2-related factor (NFE2L2) is the major regulator of the antioxidant response element (ARE) pathway, thus defined as the NRF2/ARE system [139]. NFE2L2 is sometimes designated as NRF2 and, for this reason, is missed with nuclear respiratory factor 2 (Nrf2); see below [10]. NFE2L2/NRF2 is the crucial transcription factor in sensing oxidative stress and activating genes that regulate the production of antioxidant proteins and mitochondrial biogenesis [10,140]. However, despite the clear effects of NFEL2 deficiency on oxidative stress, NFE2L2 signaling is not the only pathway controlling oxidative stress. Many protective antioxidant enzymes are regulated by several other factors, including activator protein 1 (AP-1) [141]. 

Under basal conditions, the cytosolic Keap1 targets the NFE2L2 protein for degradation in the cytosol via the ubiquitin–proteasome system. In conditions of oxidative stress, a change in Keap1 structure prevents the degradation of NFE2L2 by UPS; thus, synthesized NFE2L2 is no longer degraded and accumulates in the cytosol. Then accumulated NFE2L2 is phosphorylated by AMPK, released from the complex, and translocated to the nucleus. There. NFE2L2 binds to the antioxidant response element (ARE), which triggers the expression of more than 200 genes with AREs. NFEL2L2 promotes the transcription of various cytoprotective genes, including antioxidant enzymes such as heme oxygenase-1 (HO-1), superoxide dismutase-2 (SOD2), and others [10]. NFE2L2 also activates proteasomes (a group of two or more associated polypeptide chains that degrade damaged or unneeded proteins by proteolysis), thus regulating the quality control of proteins by removing misfolded and ubiquitinated proteins. Moreover, NFE2L2 has an influence on the expression of PINK1, so it regulates mitophagy directly and protects mitochondria from oxidative injury [10]. Direct and efficacious influence on the mitophagy and cellular senescence pathways is one of many strategies for targeted interventions in AMD, so restraining PINK1 deficiency may be one of such new healing approaches [142].

NFE2L2 positively regulates the expression gene of p62/SQSTM1 due to the presence of AREs in its promoter region, implying a positive feedback loop. On the other hand, p62/SQSTM1 promotes autophagic degradation of Keap1 and activation of the NFE2L2 gene response. It forms another positive feedback loop, i.e., activation of NFE2L2 in response to oxidative stress via autophagy-related p62/SQSTM1 [10]. 

During AMD development, many cytoprotective mechanisms are first upregulated, including antioxidant production and autophagy machinery. With time, when the disease shifts to a chronic state, the antioxidative capacity diminishes, the mitophagy process becomes sluggish, and cytoprotection decreases [10]. The spontaneous decline of the NFE2L2/ARE pathway and increased oxidative stress during aging call for functional damage-limiting pathways, including autophagy and the inflammatory response [10,143]. The results of laboratory research performed in mouse models suggest that the NFE2L2/ARE signaling cascade is the most promising target for pharmacological intervention in AMD since NFE2L2 participates in the maintenance of mitochondrial homeostasis and NFE2L2 activators are capable of modulating the processes of mitochondrial biogenesis and mitophagy [10].

The peroxisome proliferator-activated receptor gamma coactivator-1 (PGC-1) family consists of PGC-1α, PGC-Ιβ, and PRC (PGC-1-related coactivator). All these proteins control the mitochondrial antioxidant defense system, mitochondrial biogenesis, and respiratory function. In response to oxidative stress, PGC-1α is phosphorylated by AMPK in the cytosol, similar to NFRL2L2 activation. Active PGC-1α translocates to the nucleus, where it triggers the transcription of many genes with AREs that regulate antioxidant protein production and mitochondrial biogenesis [10]. The processes of mitochondrial biogenesis and autophagy/mitophagy pathways can be intensified by prompting PGC-1α via silent information regulator 1 (Sirt1) [138,144]. Sirt1 phosphorylation by AMPK leads to deacetylation and activation of PGC-1α [137]. PGC-1α enhances RPE metabolic functions and resistance to oxidative stress through the induction of oxidative metabolic genes, antioxidant enzymes, and the regulation of autophagy/mitophagy [10]. A deficiency of PGC-1α protein increases the amount of stress markers characteristic of oxidative mitochondrial damage. Resveratrol has the ability to activate Sirt1 indirectly by raising NAD+ levels and stimulating AMPK [145]. PGC-1α, Sirt1, and AMPK represent new therapeutic targets for interventions in AMD disorder [138].

PGC-1α also controls the expression of the NFE2L2 gene. It suggests the existence of a feed-forward loop promoting the expression of protective genes, and such a PGC-1α/NFE2L2/ARE loop increases the activity of the proteasome and autophagy, leading to decreased protein aggregation, a characteristic of aging and neurodegenerative diseases [10].

### 3.2. Antioxidative Role of the p62/SQSTM1/Keap1-Nrf2/ARE Pathway

Nrf2, the nuclear respiratory factor 2, functions as a transcription factor, activating the expression of key metabolic genes regulating cellular growth and nuclear genes required for mitochondrial respiration, DNA transcription, and replication [10]. Nrf2 factor also acts as a great antioxidative inducer, having the ability to regulate multiple antioxidant enzymes to preserve the cell from oxidative insult [83]. Under non-stressed conditions and in the absence of oxidative stress, Keap1 maintains Nrf2 isolated (hidden) in the cytosol, where Keap1 restrains Nrf2 proteasomal degradation and deactivation [83,146]. However, upon excessive ROS exposure and oxidative stress conditions, cysteine residues of Keap1 are oxidized, and oxidized Keap1 dissociates from Nrf2. It allows Nrf2 to translocate to the nucleus. There, active Nrf2 binds to ARE in the promoter regions of its target genes and initiates the transcription of antioxidative genes [83,147]. The Nrf2 factor does not significantly regulate the essential expression of the antioxidant enzymes. The principal function of Nrf2 is to induce the early (acute) and late (chronic) phases of the antioxidant response. The early phase of the Nrf2-induced antioxidant response is mediated by the direct enzymes catalase and superoxide dismutase 2 (SO_2_). The chronic phase of the Nrf2-induced antioxidant response depends on maintaining an appropriate amount of glutathione S-transferase (GSH) enzymes, thioredoxin (Trx)-generating enzymes, and heme oxygenase-1 (HO-1) enzymes. Nrf2 plays a role not only as an inducer of antioxidant responses. but also as an important cell survival factor since GSH deficiency correlates directly with oxidative cell injury and cell death via the apoptotic mechanism [146,148]. However, in aging RPE cells impaired by oxidative stress, Nrf2 signaling fails to activate [149]. The Keap1-Nrf2/ARE pathway comprehensively regulates the protective antioxidant response; it is involved in RPE cell survival responses and the initiation of retinal neuroprotection [1,83]. Upregulated Nrf2 signaling has possibilities for maintaining cellular redox homeostasis since it activates a complex of cellular antioxidant responses and connects with the antioxidative macroautophagic lysosomal cleaning pathway [1]. A laboratory study confirmed the antioxidative role of the Nrf2 factor. The Nrf2 knockout (Nrf2 KO) mice displayed symptoms of imbalance in cellular homeostasis and many hallmarks of the early and late phases of AMD pathology, i.e., SD deposits, RPE deficiency and hypopigmentation, lipofuscin autofluorescence, BrM thickening, and symptoms of GA/AMD. CNV was found in nearly 20% of the examined eyes. The Nrf2 KO mice also displayed the compromised RPE cell’s ability to phagocytize the POS (heterophagy vacuoles were abnormal, and enlarged mitochondria were close to the aberrant phagocytic vacuoles) [150]. 

Direct binding between p62/SQSTM1 and the Keap1 protein closely links the Nrf2 factor with macroautophagy regulated by p62/SQSTM1. Keap1, a sequester and inhibitor of the Nrf2 factor in the cytoplasm, normally enables the transcription of Nrf2. However, along with p62-mediated macroautophagy, p62 bounds and inactivates the Keap1 protein. The active Nrf2 factor stimulates transcription of the antioxidant genes to protect against oxidative stress and promotes autophagic degradation of Keap1 [151]. p62/SQSTM1 enhances Nrf2 activity, and Nrf2 upregulates p62 expression at the transcriptional level, forming a positive feedback loop [152]. 

Understanding the regulatory mechanisms that control Nrf2 protein levels, along with the molecular mechanisms of UPS and autophagy, will guide the future development of Nrf2-targeted therapeutics in AMD [1].

### 3.3. Antioxidative Role of Circulating MicroRNAs

MicroRNAs (miRNAs) are a class of endogenously expressed, single-stranded, small (containing 17–25 nucleotides), and non-coding RNA molecules; these functional molecules are not translated into proteins. MiRNAs are located within introns and exons of protein-coding genes or in intergenic regions and function by silencing post-transcriptional gene expression. MiRNAs are involved in gene silencing by repressing protein synthesis through imperfect binding to the 3’-UTR (untranslated region) of the target messenger RNA (mRNA), which leads to mRNA degradation. MiRNAs play a crucial role in the regulation of gene expression and participate in many physiologic signaling networks and biological processes (cell proliferation, differentiation, migration, growth control, organogenesis, apoptosis, and angiogenesis). Mature miRNAs are released into the extracellular fluid and transported to target cells directly by binding to proteins or indirectly via exosomes (membrane-bound extracellular vesicles). Their disruption or malfunction leads to many general disorders (cancer, inflammation, diabetes, cardiovascular, and neurologic diseases) [153,154]. Altered expressions (dysregulations) of different miRNAs are involved in AMD etiopathogenesis [155,156,157]. Circulating miRNAs may be potential diagnostic biomarkers for AMD [158,159]. Alterations in miRNA-9, miRNA-21, miRNA-23a, miRNA-27a, miRNA-126, miRNA-144, miRNA-146, miRNA-150, miRNA-155, and Let-7 are associated with both advanced AMD forms [156]. According to another study, miRNA-23, miRNA-27a, miRNA-34, miRNA-126, miRNA-146a, and miRNA-155 are the most frequently dysregulated miRNAs in AMD. All of them are associated with oxidative stress, apoptosis, inflammation, and angiogenesis; however, in oxidative stress, miRNA-27a and miRNA-146a are mainly involved; miRNA-27a affects RPE by prooxidants, and miRNA-146a is up-regulated by ROS [155]. 

MiRNAs have dual countenance in the aspect of oxidative stress in RPE cells. Dysregulated miRNAs can affect the cellular redox status through the activity of enzymatic antioxidants, including miRNA-21 influence on SOD, miRNA-30b and miRNA-146a influence on catalase, and miRNA-144 and miRNA-214 influence on GPx [156]. However, miRNA-23a, miRNA-125b, miRNA-141, and miRNA-626 may exert a protective influence and attenuate RPE oxidative damage [160,161,162]. 

According to Lin et al., miR-23a expression is significantly downregulated in macular RPE cells from human AMD eyes. In his study, H_2_O_2_-induced ARPE-19 cell death and apoptosis were increased by miR-23a inhibitor and decreased by miR-23a mimic; however, forced overexpression of miR-23a decreased H_2_O_2_ or t-butylhydroperoxide (tBH)-induced Fas receptor upregulation, and this effect was blocked by downregulation of miR-23a. According to the mentioned authors, the protection of RPE cells against oxidative damage is afforded by miR-23a through regulation of Fas (death of the Fas receptor on the cell surface leads to programmed cell death, apoptosis, if it binds to Fas ligand). Fas is a functional target gene of miR-23a, which is involved in miR-23a-mediated protective effects on RPE cell injury elicited by H_2_O_2_, and it may be a novel therapeutic target in retinal degenerative diseases [160]. 

An experimental study showed that overexpression of miRNA-125b can inhibit Keap1 expression and enhance Nrf2 expression. Activation of the Nrf2 signal pathway can protect RPE cells from oxidative damage. Moreover, forced expression of miR-125b can significantly increase SOD levels while reducing ROS overproduction and malondialdehyde (MDA) formation. However, miRNA-125b has no effect when Nrf2 is silenced in ARPE-19 cells [161]. miR-125b can protect RPE cells from oxidative damage by activating the Nrf2/HIF-1α signaling cascade. The HIF1 family, composed of constitutively expressed HIF-1β and oxygen-sensitive HIF-1α, is the major transcription factor regulating the cellular response to oxygen level changes. In normoxia, HIF-1α is hydroxylated by the prolyl hydroxylase domain protein (PHD). It allows recognition of HIF-α by the Von Hippel-Lindau protein (VHL), a ubiquitin E3 ligase, which results in polyubiquitination of HIF-α and proteasome-mediated degradation. Under hypoxic conditions, PHD is inactive and the hydroxylation level of HIF-1α decreases. As a result, HIF-1α cannot be recognized by VHL, which means a lower ubiquitylation level, and thus HIF-1α accumulates and is translocated to the nucleus to exert its transcriptional activities. In the nucleus, HIF-1α binds to the hypoxia-responsive elements (HREs) on the promoter of its downstream genes and initiates transcription of hypoxia-inducible genes and the antioxidative response [162]. miR-125b can inhibit Keap1 3′-UTR activity and promote HIF-1α protein degradation to activate the Nrf2/HIF-1α signaling cascade in RPE cells and therefore protect RPE from oxidative injury [161].

miRNA-141 attenuates UV-induced oxidative stress via activating Keap1-Nrf2 signaling in human RPE cells and retinal ganglion cells [163]. Targeting Keap1 by miRNA-626 protects RPE cells from oxidative injury by activating Nrf2 signaling [164].

Circulating miRNAs in the human peripheral circulatory system have potential as diagnostic, prognostic, and/or predictive biomarkers in AMD pathology since they are present at relatively stable levels in human plasma and serum; miRNA levels have been reported to be much more stable than mRNA, and miRNAs are measurable with much greater sensitivity than proteins. However, the profiling of miRNAs in human body fluids and the analysis of miRNA panels that target ROS, repair of oxidative damage in DNA, and inflammation require the common availability of clinically useful molecular diagnostic tests [156].

### 3.4. Antioxidative Role of Yttrium Oxide Nanoparticles

A laboratory study performed on murine light-stress models with albino mice (Balb/C) showed that yttrium oxide (Y_2_O_3_) nanoparticles (NPs) act as free radical scavengers. Intravitreal injections of Y_2_O_3_ NPs (~10–14 nm in diameter) at doses ranging from 1.0 to 5.0 µM and performed 2 weeks before acute light stress ameliorated retinal oxidative stress-associated degeneration and protected photoreceptors from degeneration in both structural and functional aspects (decreased light-associated thinning of the outer nuclear layer and preservation of scotopic and photopic electroretinogram amplitudes, respectively). Higher doses were more effective, but doses less than 1.0 µM were not effective. Y_2_O_3_ NP delivery after light exposure was not effective, probably because the particles needed more time to penetrate through the vitreous and retina. The Y_2_O_3_ NPs were nontoxic and well tolerated after injection [165]. 

According to the authors, delivery of highly efficacious, non-enzymatic antioxidants directly to the eye, while avoiding issues of oral bioavailability, may significantly ameliorate retinal degeneration. Y_2_O_3_ NPs have antioxidant benefits and may be a strategy in the future for the treatment of multiple forms of retinal degeneration associated with oxidative stress [165]. 

## 4. Conclusions

Heterophagy, autophagy, and mitophagy—physiological, internal molecular pathways—act as great antioxidative and protective mechanisms in the RPE cells. Strong and prolonged (multi-annual) oxidative stress disrupts the normal antioxidative potential of RPE.

Sluggish and dysregulated with time, heterophagy, autophagy, and mitophagy mechanisms are strong contributors to AMD development in the aged RPE cells of predisposed patients.

Dynamic modeling of heterophagy, autophagy, and mitophagy machinery, together with an increase in their activity and efficiency, represents one of the therapeutic options and is also currently one of the newest and most promising strategies for target interventions in AMD disease.

Pharmacological and/or genetic modulation of mTORC1 function, which is the major negative regulator of autophagy, is a potential target for the development of therapeutics for neurodegenerative diseases and AMD. A suppression deficiency of the mitophagy regulator PINK1 is a promising target for pharmacological intervention in AMD. AMPK, NFE2L2, PGC-1α, Sirt1, and Nrf2 proteins, as well as the NFE2L2/PGC-1α/ARE signaling cascade, p62/SQSTM1/Keap1-Nrf2/ARE pathway, and circulating miRNAs, also represent new feasible therapeutic targets for interventions in AMD disorder.

## Figures and Tables

**Figure 1 antioxidants-12-01368-f001:**
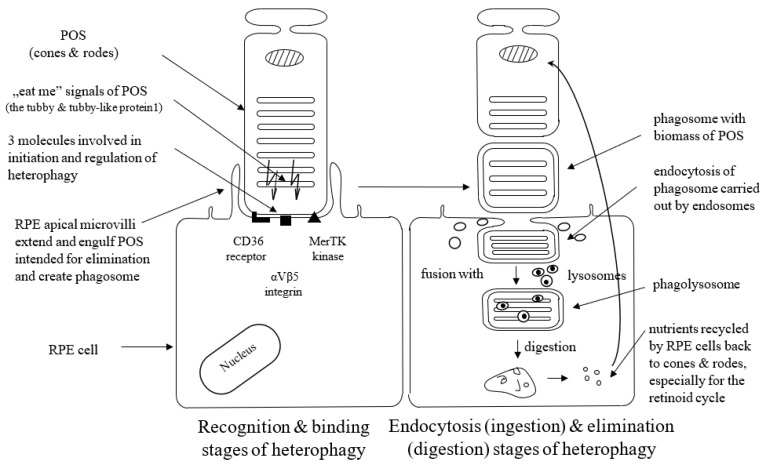
Schematic presentation of binding, ingestion, and digestion stages of heterophagy process. POS; photoreceptors’ outer segments, MerTK; tyrosine-protein kinase c-Mer; RPE; retinal pigment epithelium.

**Figure 2 antioxidants-12-01368-f002:**
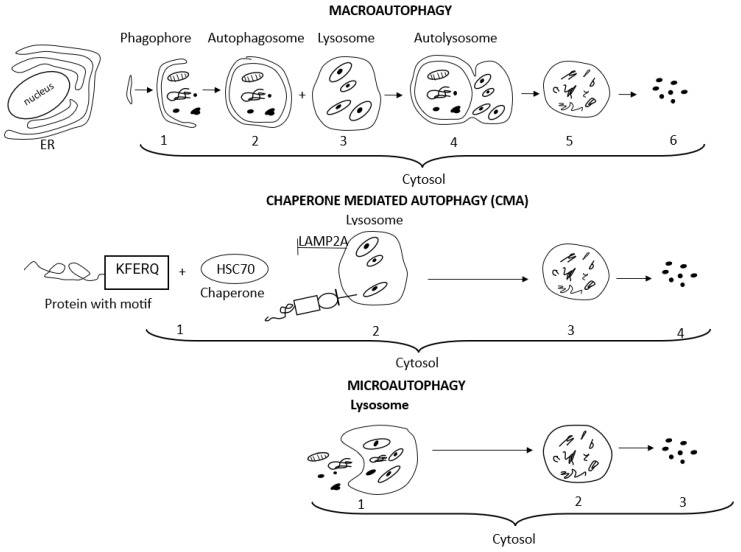
Schematic presentation of macroautophagy, chaperone-mediated autophagy, and microautophagy processes, which differ in how the targeted cytosolic content reaches the lysosomes for degradation. With regard to macroautophagy: (1) membrane isolation, vesicle nucleation, elongation, and phagophore (pre-autophagosomal structure) formation in cytosol, adjacent to the ER; (2) double-membrane autophagosome completed and matured; (3) lysosomes filled with hydrolytic enzymes; (4) vesicle breakdown, fusion of autophagosome with lysosome and autophagolysosome formation; (5) cargo degradation; (6) recycling of metabolites and nutrients to restore the nutrient balance. With regard to chaperone-mediated autophagy: (1) CMA substrates containing KFERQ-like motif are recognized by the HSC70 protein (chaperone) in the cytosol; (2) CMA substrates bind to LAMP2A; lysosome-associated membrane protein 2A. LAMP2A forms a translocation cluster through which the CMA substrates are translocated into the lysosomal lumen; (3) cargo degradation; (4) recycling of metabolites and nutrients. With regard to microautophagy: (1) direct engulfing of the cargo by invaginated lysosomal membrane without forming autophagosome; (2) cargo degradation; (3) recycling of metabolites and nutrients.

**Figure 3 antioxidants-12-01368-f003:**
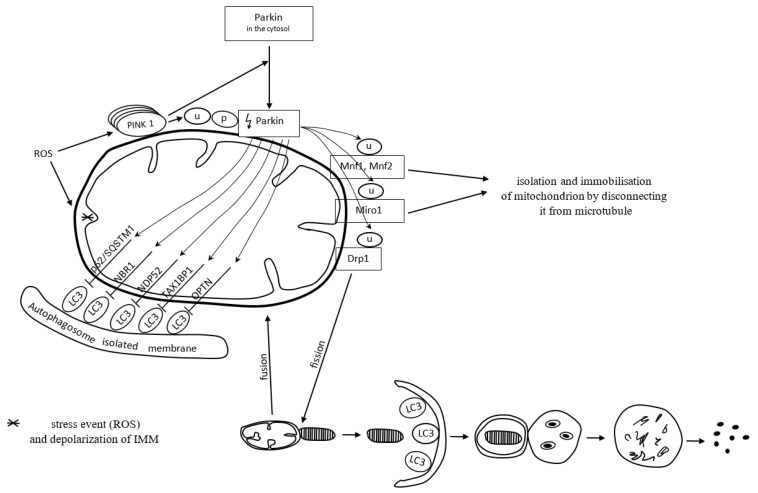
Reactive oxygen species (ROS) depolarize the inner mitochondrial membrane (IMM) and activate PINK1 Parkin-derived pathway of mitophagy. In dysfunctional mitochondria, with a loss of IMM potential, PINK1 degradation is inhibited. Accumulation of PINK1 triggers recruitment of cytoplasmic Parkin. Parkin recruited to the mitochondrial surface is ubiquitinated (U) and phosphorylated (P) by PINK1. Then, active Parkin initiates the ubiquitination of mitofusin-1 (Mfn1), mitofusin-2 (Mfn2), Miro 1, and dynamin-related protein 1 (Drp1). Inhibited activity of Mfn1, Mfn2, and Miro1 proteins facilitates disconnection of mitochondria from the microtubule. Active Drp1 results in fission (segregation) of damaged mitochondrion, which is finally escorted to the autophagic machinery for degradation. Mitophagy receptors p62/SQSTM1 (sequestosome-1), NBR1 (neighbor of BRCA1 gene 1 protein), NDP52 (nuclear domain 10 protein 52), TAX1BP1 (Tax1-binding protein 1) and OPTN (optineurin) contain a ubiquitin-binding domain, which allows their attachment to LC3 (microtubule-associated proteins 1A/1B light chain 3). The LC3 adapter is present on the surface of a maturing autophagosome. When a damaged mitochondrion is attached to an autophagosome, the autophagosome membrane elongates and closes the mitochondrion inside for lysis.

**Figure 4 antioxidants-12-01368-f004:**
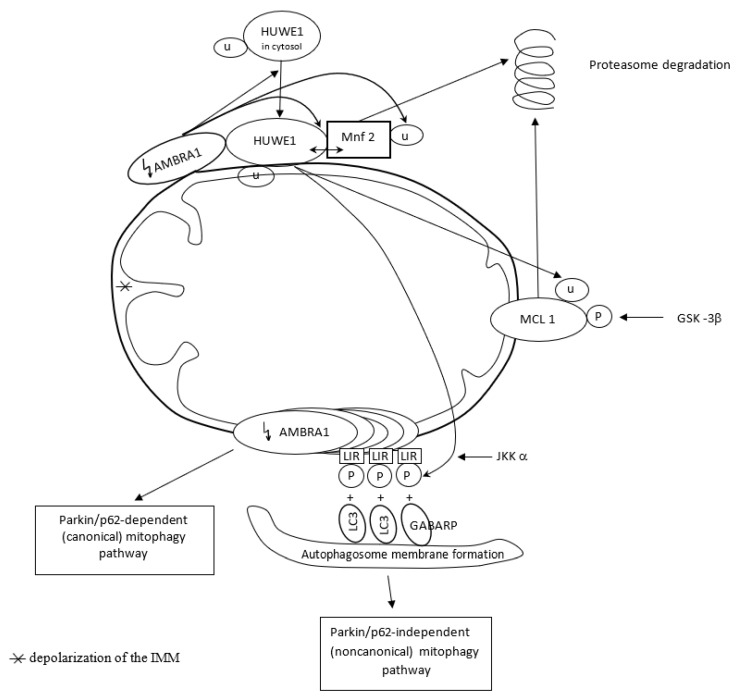
AMBRA1 acts as activator of Parkin/p62-dependent (canonical) pathway in basal conditions. However, AMBRA1 accumulation on OMM (outer mitochondrial membrane) changes mitophagy process in Parkin/p62-independent manner since AMBRA1 has the possibility to bind to the LC3 and GABARP adapter through an LIR motif, regardless of Parkin/p62 recruitment. AMBRA1 interacts with HUWE1 ligase, and such collaboration regulates mitochondrial clearance in two steps. AMBRA1 favors HUWE1 translocation from the cytosol to OMM and HUWE1 binding to Mnf2. HUWE1–Mnf2 interaction leads to Mfn2 ubiquitylation (U) and its targeting to the proteasome for enzymatic degradation. In the second step, HUWE1 induces phosphorylation (P) of LIR motif of AMBRA1 receptor, mediated by IKKα kinase, and triggers Parkin/p62-independent mitophagy.

## Data Availability

Not applicable.

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
