# Peer review of "Antioxidative Role of Heterophagy, Autophagy, and Mitophagy in the Retina and Their Association with the Age-Related Macular Degeneration (AMD) Etiopathogenesis"

_antioxidants, 2023, doi:10.3390/antiox12071368_

Round 1

Reviewer 1 Report

The review by M Nita and A Gryzybowski provides an overview of the antioxidative role of heterophagy, autophagy and mitophagy in maintaining homeostasis of the retina. They highlighted these processes as potential therapeutic approaches for Age-related macular degeneration (AMD) which is an oxidative stress-linked neurodegenerative disease. Altogether this review is of interest for the scientific community. However, I have some comments that need to be addressed:

It would really be appreciated to add figures that may describe:

- Heterophagy (3 stages such as binding, ingestion and digestion)

- The three types of autophagy with a focus on macroautophagy

- Mitophagy in mammals and next a particular focus on what is has been described about mitophagy only in the context retina (Ie: which mitophagic receptor are really involved in retina? )

Page 7: When the authors describe CMA, they should cite the original article that highlights the discovery of CMA. Infact, citations of important articles from the group of Anna maria Cuervo are missing.

Page 7: When the author discussed the origin of the autophagosome , they proposed that ER may participate in this process but after they said that it is not demonstrated. Instead, we now know that ER membranes are involved in autophagosome formation and also other sources of membranes such as plasma membranes.  This point should be better described and articles such as PMID: 34260019 , PMID: 28754694  PMID: 34643464  PMID: 27758042 etc…should be cited.

Page 11: Last sentence: Mitophgy needs to be replaced by Mitophagy

Page 12: There is a mistake. AMBRA1 does not contain an Ubiquitin-binding domain such as OPTN or NDP52. Instead, AMBRA1 should be described after, in the section in which FUNDC1 and NIX are described. Articles related to this should also be cited such as : PMID: 25215947  PMID: 30217973 PMID: 31434979.

Reviewer 2 Report

The authors tried to demonstrate the antioxidative roles of heterophagy, autophagy and mitophagy in AMD etiopathogenesis by review recently published literatures. The authors have covered large quantity of publications and provide reasonably insight for readers to understand the oxidative stress induced damage and consequence and resolution of the damage by the mechanisms involved in the review in the RPE cells. Since this is to be published in the section of the health outcomes of the antioxidants and oxidative stress, it is very important for authors to note that, by definition, heterophagy, autophagy and mitophagy can not play a direct antioxidative role(s), as antioxidant, per se. The RPE cells are normally consider as postmitotic cells, therefore any damage, including oxidative stress induced damage has to be resolved before cells are committed to one of the forms of cell death. Meanwhile, there are evidences that oxidative stress itself will cause the change of the machinery of the heterophagy, autophagy or mitophagy resulting the loss of function of those cellular homeostasis mechanisms. Whether those changes are reversible and if they make a great impact on the development of AMD need to be well illustrated.

English in general is OK, but there is a need to polish it.
